# Joint Tumor Bud–MMP/TIMP Count at the Invasive Front Improves the Prognostic Evaluation of Invasive Breast Carcinoma

**DOI:** 10.3390/biomedicines9020196

**Published:** 2021-02-16

**Authors:** Luis O. González, Noemi Eiro, María Fraile, Rosario Sánchez, Alejandro Andicoechea, Silvia Fernández-Francos, Jose Schneider, Francisco J. Vizoso

**Affiliations:** 1Department of Anatomical Pathology, Fundación Hospital de Jove, 33290 Gijón, Spain; a.patologica2@hospitaldejove.com; 2Research Unit, Fundación Hospital de Jove, 33290 Gijón, Spain; noemi.eiro@gmail.com (N.E.); maria.fraile82@gmail.com (M.F.); silviafernandezfrancos@gmail.com (S.F.-F.); 3Department of Surgery, Fundación Hospital de Jove, 33290 Gijón, Spain; charimeras@hotmail.com (R.S.); aandiko1@gmail.com (A.A.); 4Department of Obstetrics and Gynecology, University of Valladolid, 47005 Valladolid, Spain; jose.schneider@urjc.es

**Keywords:** MMPs, TIMPs, tumor invasion, metastasis, epithelial-mesenchymal transition

## Abstract

Background: Tumor budding is a histological phenomenon consisting of the formation of small clusters of one to five undifferentiated malignant cells detached from the main tumor mass which are observed in the tumor stroma. In the present study, we investigated the prognostic significance of tumor budding in breast cancer and its relationship with the expressions of matrix metalloproteases (MMPs) and their tissue inhibitors (TIMPs). Methods: The number of buds was counted in whole-tissue sections from 153 patients with invasive ductal carcinomas who underwent a long follow-up period. In addition, an immunohistochemical study of MMP-9, -11, and -14 TIMP-1 and -2 expression by cell types at the invasive tumor front was carried out. Results: There was a wide variability in the number of buds among tumors, ranging from 0 to 28 (median = 5). Tumor budding count ≥ 4 was the optimal cut-off to predict both relapse-free and overall survival. High-grade tumor budding was associated with MMP/TIMP expression by cancer-associated fibroblasts. In addition, we found that the combination of tumor budding grade with MMP/TIMP expression by stromal cells, and especially with MMP-11 expression by mononuclear inflammatory cells, significantly improved the prognostic evaluation. Conclusion: High-grade tumor budding is associated with a more aggressive tumor phenotype, which, combined with MMP/TIMP expression by stromal cells at the invasive front of the tumor, identifies patients with poor prognosis.

## 1. Introduction

From a histological and molecular point of view, breast cancer is a heterogeneous disease, with around 30% of patients developing metastasis [1]. Although molecular tests are being used to assess the likelihood of treatment response and/or recurrence, additional prognosticators are needed to enhance personalized treatment and, especially, to overcome the over- and undertreatment of patients. Tumor budding is a morphological phenomenon found in various cancers. This histological finding consists of single or small clusters of one to five undifferentiated malignant cells detached from the main tumor mass, which are seen in the tumor stroma in close proximity ahead of the invasive front of a tumor (reviewed in by the authors of [2,3]). The method of scoring tumor budding, based in hematoxylin and/or eosin-stained sections and/or pancytokeratin staining, is a single and low-cost method with high reproductivity [4].

Colorectal cancer was the first cancer type in which tumor budding was addressed systematically. In fact, tumor budding is an additional prognostic factor for this tumor type according to the Union for International Cancer Control (UICC) [5], and a potential prognostic factor in its early stage according to the European Society for Medical Oncology consensus guidelines [6]. More recently, some of these associations were also found in esophageal [7], laryngeal [8], tongue squamous cell carcinoma [9], gingival buccal complex squamous cell carcinoma [10], and lung [11], pancreatic [12], bladder [13], and gastric carcinomas [14]. In invasive ductal breast cancer, high budding has been correlated with clinical-pathological parameters such as larger tumor size [15], lymph vessel invasion [15,16,17,18], and nodal metastasis [16,17,18], as well as shorter survival [15,17,18]. However, the definitive implementation of tumor budding into clinical practice of breast cancer is currently limited by the vast heterogeneity in its exact definition, methodology of assessment, and patient stratification. Therefore, in order to better characterized this histological finding, it may be relevant to integrate tumor budding in its biological context as part of the invasive tumor front. Interestingly, it has been suggested that cells associated with tumor budding share similar properties with malignant stem cells [19], as well as a manifestation of one hybrid epithelial/mesenchymal phenotype displaying collective cell migration [3]. Apart from this last event, disruption of the barriers is a prerequisite for invasion in breast cancer, and the matrix metalloproteases (MMPs) play a key role in this process [20]. In fact, they have been clinically associated with metastasis development in many tumors, including breast cancer.

In the present study, we investigated the prognostic signification of tumor budding grade in patients with breast cancer who underwent a long follow-up period. In addition, we evaluated the relationship of this histological finding with the expression of MMPs and their tissue inhibitors (TIMPs) at the invasive front of the tumors in order to improve the prognostic evaluation of breast cancer based just on the morphological context.

## 2. Materials and Methods

### 2.1. Patients

This study comprised 153 women with a histologically confirmed diagnosis of invasive ductal breast cancer treated between 1990 and 2001, some of which were previously included in our preliminary studies on the expression of MMPs and TIMPs in breast cancer [21,22,23,24]. We selected women with the following inclusion criteria: Invasive ductal carcinoma, at least 6 histopathologically assessed axillary lymph nodes, and a minimum of 10 years of follow-up without tumor recurrence. The exclusion criteria were the following: Metastatic disease at presentation, prior history of any type of malignant tumor, bilateral breast cancer at presentation, having received any type of neoadjuvant therapy, the development of locoregional recurrence during the follow-up period, or the development of a second primary cancer. We randomly selected a sample size of 153 patients, in accordance with 4 different groups of similar size, stratified with regard to nodal status and the development or not of metastatic disease. This was in order to include a sufficient number of patients with recurrence for securing the statistical power of the survival analysis. Note that approximately half of the cases with distant metastasis during the follow-up period occurred in each of the node-negative and node-positive subgroups. Patients’ characteristics included in the 2 main groups, with or without distant metastases (recurrence), are listed in Table 1. Patients underwent either modified radical mastectomy or wide resection with axillary lymphadenectomy. The median follow-up period was about 187 months in patients without metastasis and 52 months in patients with metastatic disease. The study adhered to national regulations and was approved by our Institution’s Ethics and Research Committee. Women were treated according to the guidelines used in our Institution (Fundación Hospital de Jove). Written informed consent was obtained from all patients and controls. 

### 2.2. Definition of Tumor Budding 

Tumor buds are defined as a small number of cells (up to 5) which have detached from the bulk of the tumor and are observed as isolated cells or small clusters of cells in histologic sections. In this study, we analyzed peritumoral buds defined as those observed in areas near the margin of the tumor at the invasive tumor front [25]. The number of buds was counted in whole-tissue sections of surgical resection specimens. Prior to the selection of the areas where the count was carried out, a scan of the entire peritumoral area was made, and the areas with the highest budding density were chosen. The number of buds was counted in the hematoxylin and eosin-stained sections with the maximal invasive region and in areas with the highest concentration of buds (‘‘hotspots’’), as considered by other authors [15,16,17,18]. The areas with the highest budding density were chosen, but if 1 case had fewer high-power fields (HPFs: 0.55 mm) with any buds, the HPFs without buds (*n* = 0) were chosen to achieve 10 evaluated HPFs. The number of buds was highly variable along the invasive front, and consequently, the final number obtained was the average of the areas with the highest budding density [26]. Pan-cytokeratin immunostaining (antibody: AE1/AE3) was carried out in exceptional cases when it was difficult to distinguish budded tumor cells from fibroblasts or inflammatory cells on morphological criteria. We used predefined criteria for the assessment of tumor buds in accordance with recent publications describing the scoring of tumor buds in colon cancer [26,27]. A breast pathologist (LOG) scored 10 high-power fields (0.55 mm) blindly without knowledge of tumor characteristics or the results of the present studies. In a recent report [15], tumors were considered to have a high tumor budding if the average number of tumor buds in 10 HPF was > 4. In contrast, tumors were considered to have a low tumor budding if the average number of buds in 10 HPF was ≤ 4. Figure 1 shows representative examples of either low or high tumor budding.

### 2.3. Tissue Arrays 

Breast carcinoma tissue samples were obtained at the time of surgery. Tumor tissue array (TA) blocks were obtained by punching a tissue cylinder (core) with a diameter of 1.5 mm through a histologically representative area of each ‘donor’ tumor block (routinely fixed (overnight in 10% buffered formalin), paraffin-embedded tumor samples), using a manual tissue arrayer (Beecker Instruments, Sun Praerie, WI, USA) as described elsewhere [22]. A total of 2 cores was employed for each case, which corresponded to the invasive front and have been shown to correlate well with conventional immunohistochemical staining [22]. The invasive front was defined as the area within 2 mm surrounding the tumors which contained cancerous cells. 

### 2.4. Immunohistochemistry

Histopathologically representative tumor areas were defined in hematoxylin- and eosin-stained sections and marked on the slide. TA blocks were obtained from areas of nonnecrotic cancerous tissues. Serial 5 µm sections of the high-density TA blocks were consecutively cut with a microtome (Leica Microsystems GmbH, Wetzlar, Germany) and transferred to adhesive-coated slides. One section of each tissue array block was stained with H&E, and these slides were then reviewed to confirm that the sample was representative of the original tumor area. Immunohistochemical staining was carried out on these sections using a TechMate TM50 autostainer (Dako, Glostrup, Denmark). Antibodies for MMPs and TIMPs were obtained from Neomarker (Lab Vision Corporation, Fremont, CA, USA). The dilution for each antibody was established based on negative and positive controls (1/50 for MMP-14 and TIMP-2, 1/100 for MMP-9 and TIMP-1, and 1/200 for MMP-11). The negative control was DakoCytomation mouse serum diluted to the same mouse IgG concentration as the primary antibody. All the dilutions were made in Antibody Diluent (Dako) and incubated for 30 min at room temperature. In a prior report, we confirmed the presence of the evaluated proteins by Western blot analysis of breast tumor cytosol samples. A single band of the expected molecular mass was observed for each protein [28]. We also used other antibodies for several factors, such as MMP-11 (clone SC3-05, 1/100, Calbiochem MERCK KgaA Darmstadt Germany). On the other hand, we also used antibodies against cytokeratins (AE1–AE3, DAKO 1/1) and vimentin (DAKO 1/100) to distinguish fibroblasts from tumor cells and CD45 (leukocyte common antigen, LCA) for macrophages. Tissue sections were deparaffinized in xylene and then rehydrated in graded concentrations of ethyl alcohol (100%, 96%, 80%, 70%, then water). To enhance antigen retrieval only for some antibodies, TA sections were microwave-treated in a H2800 Microwave Processor (EBSciences, East Granby, CT, USA) in citrate buffer (Target Retrieval Solution; Dako) at 99 °C for 16 min. Endogenous peroxidase activity was blocked by incubating the slides in peroxidase-blocking solution (Dako) for 5 min. The EnVision Detection Kit (Dako) was used as the reactivity detection system. Sections were counterstained with hematoxylin, dehydrated with ethanol, and permanently cover-slipped. We evaluated the immunohistochemical staining in the whole-tissue array section for each main cell type: Cancer cells; mononuclear inflammatory cells, including lymphocytes, plasma cells, and monocytes/macrophages (MICs); and cancer-associated fibroblasts (CAFs). We distinguished stromal cells from cancer cells because these latter cells are larger in size. In addition, fibroblasts are spindle cells, whereas mononuclear inflammatory cells are round cells. On the other hand, while cancer cells are arranged to form either an acinar or a trabecular pattern, stromal cells are isolated. Moreover, we used 2 markers to distinguish fibroblasts from tumor cells, cytokeratins and vimentin, as described above.

### 2.5. Statistical Analysis

The Chi-Square test was used to calculate significant differences between categorical variables. *p* < 0.05 was considered significant. For metastasis-free survival analysis, we used Cox’s univariate method. Cox’s regression model was used to examine interactions of different prognostic factors in a multivariate analysis, and all factors with *p* < 0.05 in the univariate analysis were included in the multivariate analysis. The PASW statistics 18 program (SPSS Inc., Chicago, IL, USA) was used for all calculations.

## 3. Results

### 3.1. Bud Quantification and Its Relationship with Clinical Outcome and Clinicopathological Features

As can be seen in Figure 2, there was a wide variability of bud number in all tumors, ranging from 0 to 28, with 5 as the median value. In general, we found that areas with high budding had a more infiltrative pattern and peritumoral stroma response.

The potential relationships between tumor budding and relapse-free survival and overall survival were evaluated in all patients included in the present study who were metastasis-free at the time of initial diagnosis. We examined all possible values obtained by counting tumor budding as cut-off points for predicting relapse-free survival. This analysis led us to define a tumor budding count of four as the optimal cut-off (X^2^ = 19.1; *p* < 0.0001) (Figure 3A). Differences in both relapse-free survival and overall survival curves were significant taking this cut-off value of four (*p* < 0.001, for both) (Figure 3B,C, respectively). In addition, Cox’s regression model demonstrated that tumor stage (stage II: RR (Relative Risk): 1.00, (CI (Confidence Interval): 0.6–1.8); stage III: 2.6 (1.3–5.1); *p* = 0.001, (SBR (Scarff-Bloom-Richardson grade) II: 2.3 (1.3–4.2); SBR III: 2.2 (1.2–4.1); *p* = 0.014; and tumor budding (2.9 (1.8–4.7); *p* < 0.0001) were independent prognostic factors.

Table 2 shows the relationship between tumor budding grade and clinicopathological features. Budding grade correlated significantly with patient age (*p* = 0.011). Thus, tumors from older women had more tumor buds that tumors from younger patients. However, our results did not show significant associations between this histological finding and other clinicopathological features, including menopausal status, tumor size, nodal status, histological grade, and ER (estrogen receptors), PgR (progesterone receptors), and HER-2 (human epidermal growth factor receptor 2) status.

### 3.2. Expression of MMPs and TIMPs at the Invasive Front and Their Relationship with Tumor Budding Grade

Considering the importance of MMPs/TIMPs in tumor progression, especially in invasion and metastasis, we also investigated their possible relationship with tumor budding grade at the invasive front of breast carcinomas in all cases included in the present study. In addition, we investigated the expressions of these factors according to each cell type. In tumors positive for cells expressing either MMPs or TIMPs, at least 70% of these cells showed a positive immunostaining of each evaluated field.

The majority of MMPs and TIMPs were mainly expressed in cancer cells at the invasive front (MMP-9: 97.3% of the tumors, MMP-11: 98.7%, MMP-14: 94.0%, TIMP-1: 86.7%, TIMP-2: 96.7%). Remarkably, as shown in Figure 4A, we found a similar MMP/TIMP expression both in cancer cells from tumor buds and those from the tumor mass at the invasive front, with differences for MMPs and TIMPs immunostaining of less than 5%. These proteins were also expressed by stromal cells but in a lower percentage of tumors. We found no significant differences for MMPs and TIMPs immunostaining between tissue array sections of each tumor or between the different areas of each section. Figure 4 shows representative examples of CAFs and MICs expressing MMPs and TIMPs localized at the invasive front in breast carcinomas. Immunostaining for these proteins revealed a cytoplasmic location in cancer cells and tumor-associated stromal cells, including both CAFs and MICs. In neoplasms positive for CAFs and MICs expressing either MMPs or TIMPs, at least 70% of these cells showed a positive immunostaining in each evaluated field. Immunostaining for these proteins had a cytoplasmic location in all positive cases. With regard to MMP-14 expression, the immunostaining showed both the cytoplasmic and membrane location. 

As can be seen in Table 3, we found several significant associations between tumor budding and MMP/TIMP expression by cell type. Considering MMP/TIMP expression, tumor budding showed a positive and significant association with MMP-11 (*p* = 0.029), MMP-14 (*p* = 0.024), and TIMP-1 (*p* = 0.021) expression by CAFs and MMP-14 expression by tumor cells (*p* = 0.034).

### 3.3. Single Combination of Tumor Budding Grade and MMP/TIMP Expression Improves Prognostic Evaluation

To assess the contribution of the joint evaluation of tumor budding and MMP/TIMP expression to the prediction of clinical outcome, we explored all possible combinations.

Our results showed several associations with a tumor budding grade, which significantly affected prognosis. Thus, high-grade tumor budding and expression by MICs of MMP-9, -11, -14, TIMP-1, or TIMP-2 were associated with a poor prognosis in all cases, whereas the association of low tumor budding count and the non-expression of either MMPs or TIMPs was associated with a better outcome (Figure 5). Similarly, we also found significant associations between tumor budding grade and the different MMP/TIMP expressions by CAFs (Figure 6).

Cox’s regression model demonstrated that tumor stage (Stage II: RR: 1.04 (CI: 0.6–1.9); stage III: 2.2 (1.1–4.6); *p* = 0.022), and tumor budding combined with MMP11 expression by MICs (low-grade/MMP11 +: 9.8 (3.3–28.8); high-grade/MMP11-: 5.9 (1.9–17.6); high-grade/MMP11 +: 20.8 (7.1–60.4) *p* < 0.0001) were independent prognostic factors.

## 4. Discussion

We selected invasive ductal breast carcinomas because they are the most prevalent histological subtype. In addition, they represent a heterogeneous group with varying tumor features and clinical outcomes in which it would be of great clinical value to find additional prognostic factors. Our results are in agreement with previous reports demonstrating that tumor budding is a morphological finding related to worse prognosis in several malignant tumors [7,8,9,10,11,12,13,14,29,30,31], including breast cancer [15,16,17,18]. In the present study, we found that tumor budding count was associated with MMP/TIMP expression, and that the combination between both factors identified a population of patients differing in their clinical outcome.

Since tumor budding at the invasive front has been postulated as the first step of invasion and metastasis [32], this histological feature could be a sensitive indicator of tumor aggressiveness. However, it has been reported that tumor cells from buds do not differ from those belonging to the tumor mass with regard to their both biological and therapeutic target factors in breast cancer, such as ER or HER-2 status [33], and even show a lower Ki67 index compared to those from the tumor mass [15]. In the present study, we found a similar MMP/TIMP expression in tumor cells from buds and those from the invasive front mass. Nevertheless, although analyzed as a histological finding represented by a tiny number of tumor cells compared with those of the tumor body [34], tumor budding is considered to reflect tumor migration in the context of the epithelial-mesenchymal transition (EMT), or at least constitutes a hybrid epithelial/mesenchymal phenotype displaying collective cell migration [3] This migration consists of epithelial cells, which lose their cell–cell adhesion and gain migratory and invasive traits that are typical of mesenchymal cells. In accordance with this biological event, it has been shown that tumor cells from buds from breast cancer show phenotypic features, such as the lack of E-cadherin expression and expression of vimentin [15,35]. However, it is also known that tumor invasion depends not only on the biological aggressiveness of tumor cells but also on the tumor microenvironment. In fact, it has been previously reported that tumor budding correlates significantly and positively with the tumor stroma percentage [17]. It has been proposed that tumor stroma has an important role in facilitating tumor cell dedifferentiation and dissemination, perhaps providing suitable energy substrate and reducing the buildup of metabolic waste [36]. A functional variability of tumor stroma cells which influences the tumor progression can also be hypothesized. Two well-studied cellular components of the tumor stroma are CAFs and MICs, both of which may be present within the invasive front, contributing to tumor progression through several biological mechanisms such as the secretion of several molecules [24,37]. One of these stromal derived molecular factors which may influence tumor aggressiveness are MMPs. It is known that these enzymes play a key role in the degradation of the extracellular matrix and basal membrane, which are prerequisites for invasion and metastasis [20]. In addition, it has also been shown that MMPs display other biological activities related to tumor progression, such as the induction of proliferation [38,39], apoptosis-resistant cells [40], neoangiogenesis [41], and the facilitation of EMT by increasing cell mobility [42,43] or by repressing the expression of E-cadherin and other cell surface adhesion molecules [44,45]. On the other hand, TIMPs do not only have an inhibitory effect against MMP actions but also are involved in protumoral activities such as the induction of proliferation and the inhibition of apoptosis [46,47].

Taken together, in the present work, we integrated a morphological feature, tumor budding grading, in the context of the biological phenotype of the stromal cells from the invasive tumor front. Our own data show that the combination between budding count and MMP/TIMP expression by stromal cells strongly improved the prognostic evaluation. Thus, whereas the concurrence of low budding count and the absence of MMP/TIMP expression by stromal cells were significantly associated with an excellent prognosis, conversely, the concurrence of high budding count with the expression of MMPs/TIMPs by these stromal cells were associated with the occurrence of distant metastasis and shortened survival. Especially relevant was our finding that the combination of tumor budding grade with MMP-11 expression by stromal MICs was significantly and independently associated with the highest prognostic accuracy. This finding points to the importance of the functional type of the tumor inflammatory infiltrate, as well as MMP-11 expression in the tumor progression of breast carcinomas. MMP-11 (also named stromalysin 3) was found to increase cancer cell survival and implantation during the early steps of the adjacent connective tissue invasion [48]. Likewise, our group previously reported that tumors with MMP-11 expression by MICs showed a high cell immune ratio ((CD68^+^) macrophages/(CD3^+^) T cells + (CD20^+^) B cells) at the invasive front [49], an upregulation of inflammatory-related genes [50,51], and a poor prognosis in breast carcinomas [24,28]. Interestingly, in a recent report by our team, we showed that the coculture of CAFs, and especially those derived from breast carcinomas showing MMP-11 expression by stromal MICs, significantly increased the invasive capability of tumor cells with the cell line MDA-MB-231 [52]. Therefore, all these data led us to consider the importance of integrating the morphological evaluation of tumor budding grade at the invasive tumor front with biological aspects of stromal cells in this relevant scenario from breast carcinomas.

This study presents some limitations. First, the study had the difficulty of evaluating the heterogeneity of counted tumor budding along the tumor sample. We suggest that further studies be carried out to explore a more accurate and rapid evaluation of this morphologic finding, such as the use of artificial intelligence tools. Second, it would be interesting to integrate an evaluation of intratumor budding, which is quantified within the main tumor mass and has been associated with clinical outcome in colorectal cancers [53,54] and observed in breast cancer [16]. Third, further studies may be of interest in other breast cancer patient populations in order to evaluate the optimal cut-off point of tumor budding count for prognosis when we combine it with MMP/TIMP expression. In this sense, further studies with patients who underwent neoadjuvant treatment may be of interest due to the fact that is the standard of care in patients with breast cancers, especially in these ones with stage III tumors. In the present study, in a population of women with primary breast cancer randomized according to distant metastasis development and a long follow-up period, we found that a value of tumor budding count of 4 (out of a range from 0 to 28) was the optimal cut-off point to predict distant relapse-free survival. Interestingly, this cut-off for tumor budding at the invasive tumor front was also associated with metastatic properties in breast cancer by other authors [16]. However, a cut-off in a similar range of seven buds per field was also considered as an optimal cut-off value to predict clinical outcome in breast cancer by other authors [15]. 

In conclusion, our data show that a high tumor budding grade is a histological feature in breast carcinomas associated with a more aggressive tumor phenotype and which may increase metastatic potential. In addition, our results enhance the clinical relevance of tumor budding at the invasive front when this morphological finding is evaluated in the context of the biological phenotype of stromal cells which contribute to tumor progression. Therefore, our data may contribute to a better prognosis interpretation of tumor budding, avoiding several issues related with reproducibility among pathologies or quantification criteria.

## Figures and Tables

**Figure 1 biomedicines-09-00196-f001:**
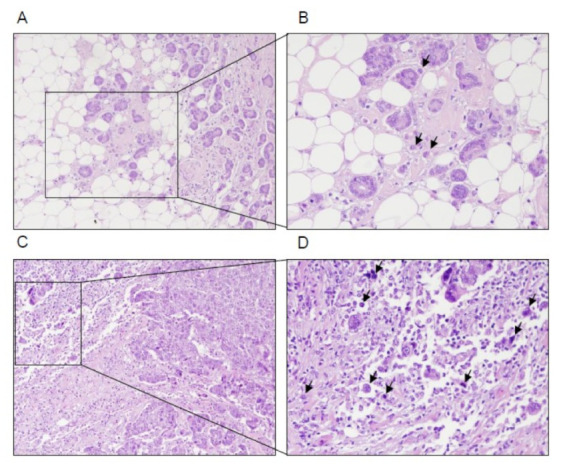
Breast cancer with low-grade tumor budding (less than 4 tumor buds). (**A**) Overview (×100) and (**B**) high-power field (×200). (**C**) Breast cancer with high-grade tumor budding (more than 4 tumor buds). (**D**) Overview (×100) and high-power field (×200). Black arrows point to tumor buds.

**Figure 2 biomedicines-09-00196-f002:**
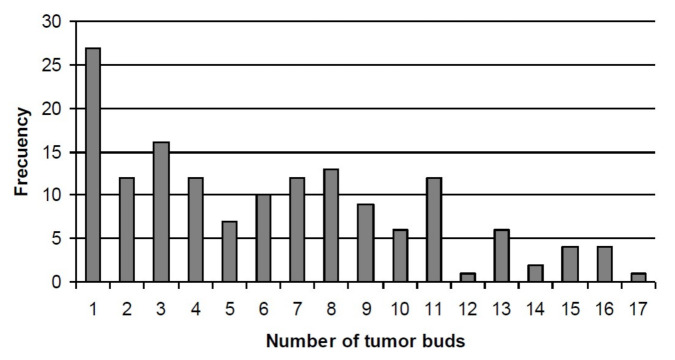
Distribution of the number of tumor buds in all tumors of 153 patients with breast cancer.

**Figure 3 biomedicines-09-00196-f003:**
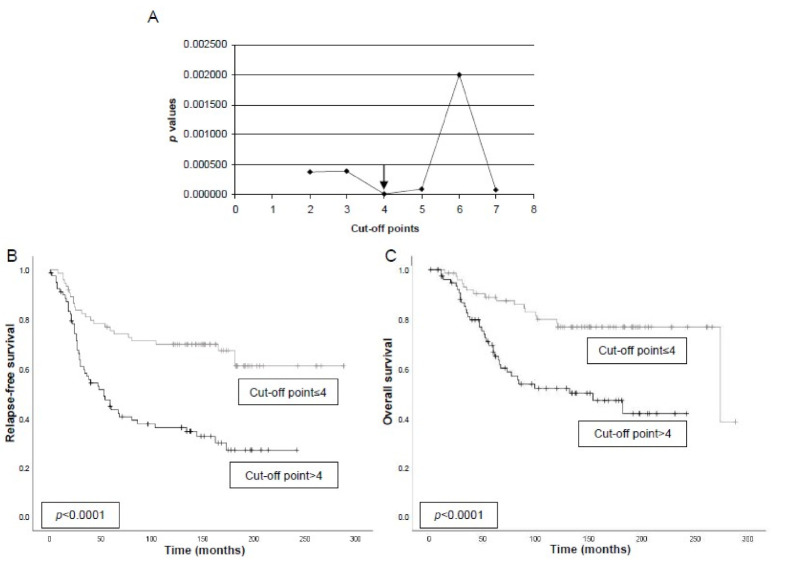
Maximum likelihood determination of the cut-off value of tumor budding count of tumors for predicting relapse-free survival in 153 patients with breast cancer. *p*-values obtained for each cut-off value are plotted against the value itself. Statistical significance is indicated by the horizontal line at the 0.05 level. (**A**) Analyses lead to the definition of a count value of four for tumor budding as the optimal cut-off (X^2^ = 19.1; *p* < 0.0001). (**B**) Relapse-free survival as a function of the tumor budding count of four as the optimal cut-off for predicting relapse-free survival. Differences in relapse-free survival curves were significant at *p* < 0.0001. (**C**) Overall survival as a function of tumor budding count of four. Differences in overall survival curves was significant at *p* < 0.0001.

**Figure 4 biomedicines-09-00196-f004:**
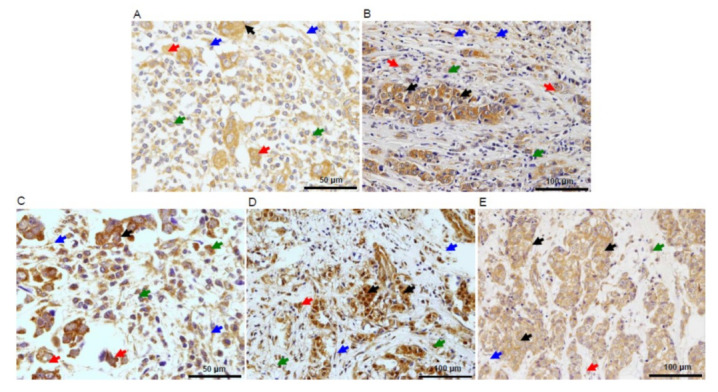
Representative examples of the expression of matrix metalloproteases (MMPs) and their tissue inhibitors (TIMPs) by stromal cells among buds at the invasive front of breast carcinomas. (**A**) MMP-9 (×400), (**B**) MMP-11 (×200), (**C**) MMP-14 (×400), (**D**) TIMP-1 (×200), and (**E**) TIMP-2 (×200). Black arrows point to cancer cells, red arrows point to tumor buds, green arrows to point mononuclear inflammatory cells (MICs), and blue arrows point to cancer-associated fibroblasts (CAFs).

**Figure 5 biomedicines-09-00196-f005:**
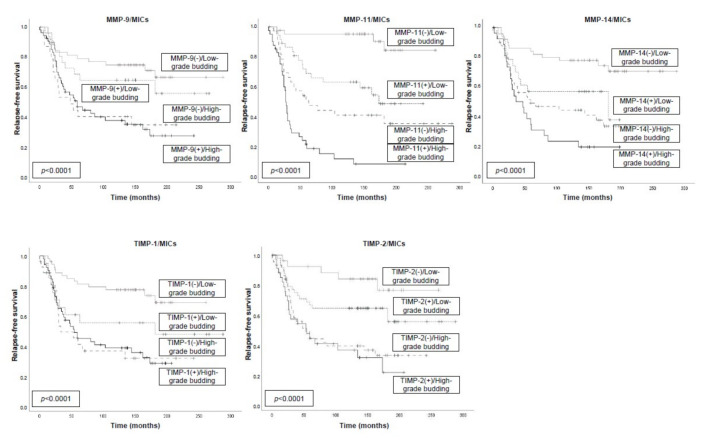
Kaplan–Meier survival curves (relapse-free survival and overall survival) as a function in 153 breast cancer patients, stratified according to tumor budding grade and the expression of MMPs and their inhibitors (TIMPs) by MICs. Tumor budding was dichotomized into low-grade (≤4) or high-grade (>4). Samples on tissue sections were insufficient or lost for analysis in three cases of MMP-9, MMP-11, TIMP-1, and TIMP-2 and in four cases of MMP-14.

**Figure 6 biomedicines-09-00196-f006:**
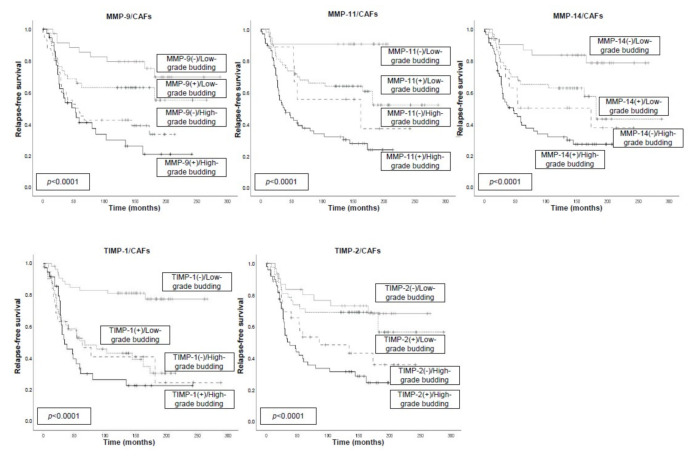
Kaplan–Meier survival curves (relapse-free survival and overall survival) as a function in 153 breast cancer patients, stratified according to tumor budding grade and the expression of MMPs and their inhibitors (TIMPs) by CAFs. Tumor budding was dichotomized into low-grade (≤4) or high-grade (>4). Samples on tissue sections were insufficient or lost for analysis in three cases of MMP-9, MMP-11, TIMP-1, and TIMP-2 and in four cases of MMP-14.

**Table 1 biomedicines-09-00196-t001:** Basal characteristics of 153 patients with invasive ductal carcinoma of the breast.

Characteristics	Without Recurrence	With Recurrence
	*N*º (%)	*N*º (%)
Total cases	76(100)	77(100)
Age (years)		
≤55	40(52.6)	40(51.9)
>55	36(47.4)	37(48.1)
Menopausal status		
Premenopausal	21(27.6)	22(28.6)
Postmenopausal	55(72.4)	55(71.4)
Tumor size		
T1	42(55.3)	30(39.0)
T2	34(44.7)	47(61.0)
Nodal status		
N(-)	34(51.5)	30(39.0)
N(+)	32(48.5)	47(61.0)
Histological grade		
Well Dif.	33(43.4)	17(22.1)
Mod. Dif.	21(27.6)	34(44.2)
Poorly Dif.	22(28.9)	26(33.8)
Estrogen receptors		
Negative	18(23.7)	32(41.6)
Positive	58(76.3)	45 (58.4)
Progesterone receptors		
Negative	22(28.9)	41(53.2)
Positive	54(71.1)	36(46.8)
Tumor stage		
I	22(28.9)	16(20.8)
II	46(60.5)	39(50.6)
III	8(10.5)	22(28.6)
HER2 status		
Negative	55(72.4)	52 (67.5)
Positive	19(25.0)	20(26.0)
Molecular types		
Luminal A	41(53.9)	26 (33.8)
Luminal B	20(26.3)	23(29.9)
HER2	4(5.3)	6(7.8)
Basal-like	9(11.8)	17(22.1)
Groups of treatment		
TMX	28(36.8)	19(24.7)
CMT	21(27.6)	28(36.4)
TMX+CMT	21(27.6)	17(22.1)
No treatment	6(7.9)	13(16.9)

HER2: Human epidermal growth factor receptor 2; TMX: Tamoxifen, CMT: Chemotherapy, TMX+CMT: Tamoxifen+Chemotherapy.

**Table 2 biomedicines-09-00196-t002:** Relationship between tumor budding grade and clinicopathological characteristics in 153 patients with invasive ductal carcinoma of the breast.

Characteristics	Low-Grade Budding (≤4)	High-Grade Budding (>4)	*p* Value
	*N*º (%)	*N*º (%)	
Total cases	74(100)	79(100)	
Age (years)			0.011
≤55	47(52.6)	33 (51.9)	
>55	27 (47.4)	46 (48.1)	
Menopausal status			0.183
Premenopausal	25(27.6)	18(28.6)	
Postmenopausal	49(72.4)	61 (71.4)	
Tumor size			0.130
T1	40(55.3)	32 (39.0)	
T2	34(44.7)	47(61.0)	
Nodal status			0.777
N(-)	32(51.5)	37(39.0)	
N(+)	42(48.5)	42(61.0)	
Histological grade			0.666
Well Dif.	25(43.4)	25(22.1)	
Mod. Dif.	24(27.6)	31(44.2)	
Poorly Dif.	25(28.9)	23(33.8)	
Estrogen receptors			0.913
Negative	25 (23.7)	25(41.6)	
Positive	49(76.3)	54 (58.4)	
Progesterone receptors			0.750
Negative	29(28.9)	34(53.2)	
Positive	45(71.1)	45(46.8)	
Tumor stage			0.139
I	23(28.9)	15(20.8)	
II	40(60.5)	45(50.6)	
III	11(10.5)	19 (28.6)	
HER2 status			0.307
Negative	56(72.4)	51 (67.5)	
Positive	16(25.0)	23(26.0)	
Molecular types			0.520
Luminal A	35(53.9)	32 (33.8)	
Luminal B	18(26.3)	25(29.9)	
HER2	4(5.3)	6(7.8)	
Basal-like	15(11.8)	11(22.1)	
Groups of treatment			0.220
TMX	23(36.8)	24 (24.7)	
CMT	26(27.6)	23(36.4)	
TMX+CMT	20(27.6)	18(22.1)	
No treatment	5(7.9)	14(16.9)	

HER2: Human epidermal growth factor receptor 2; TMX: Tamoxifen, CMT: Chemotherapy, TMX+CMT: Tamoxifen+Chemotherapy. In bold *p* value ≤ 0.05 (Chi-Square test).

**Table 3 biomedicines-09-00196-t003:** Relationship between tumor budding grade and the expression of MMPs and TIMPs by the different cell types at the invasive front of invasive ductal carcinomas of the breast.

	Low Grade			High Grade					
	Tumor Cells	MICs	CAFs	Tumor Cells	MICs	CAFs	*p* Value Tumor Cells	*p* Value MICs	*p* Value CAFs
**MMP-9**	73 (100)	25 (34.2)	40 (54.8)	73 (94.8)	24 (31.2)	40 (51.9)	0.142	0.820	0.853
**MMP-11**	71 (97.3)	32 (43.8)	52 (71.2)	77 (100)	40 (51.9)	67 (87)	0.453	0.406	0.029
**MMP-14**	65 (89)	23 (31.5)	42 (57.5)	75 (98.7)	27 (35.5)	58 (76.3)	**0.034**	0.729	0.024
**TIMP-1**	60 (82.2)	20 (27.4)	19 (26)	70 (90.9)	27 (35.1)	35 (45.5)	0.184	0.403	0.021
**TIMP-2**	70 (95.9)	47 (64.4)	44 (60.3)	75 (97.4)	43 (55.8)	49 (63.6)	0.952	0.368	0.798

Data are expressed as the number of positive cases (%). Samples on tissue sections were insufficient or lost for analysis in three cases of MMP-9, MMP-11, TIMP-1, and TIMP-2 and in four cases of MMP-14. In bold *p* value ≤ 0.05 (Chi-Square test).

## Data Availability

The data presented in this study are available on request from the corresponding author

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
