# Peer review of "Joint Tumor Bud–MMP/TIMP Count at the Invasive Front Improves the Prognostic Evaluation of Invasive Breast Carcinoma"

_biomedicines, 2021, doi:10.3390/biomedicines9020196_

Round 1

Reviewer 1 Report

Dear Authors,

presented article is interesting, but I have two comments:

  1. The collected group is not very large, especially after division into subgroups, which certainly affects the obtained results of statistical analyzes
  2. To the study group didn't include patients after neoadiuvant treatment. Actually neoadiuvant treatment is the standard of care in patient with breast cancer, especially in CSIII. Therefore I think, that, your research is of cognitive but less clinical value...

I know, that now, you aren't able to modify Yours manuscript to take into account my comments, but, maybe, they could be dicussed?

Kind regards

Author Response

Thank very much for your consideration of our work. In accordance with your suggestion, we have added a sentence in the revised version of the manuscript (Discussion section: page 13, line: 353): “In this sense, further studies with patients who underwent neo-adjuvant treatment may be of interest due to the fact that is the standard of care in patients with breast cancers, especially in these ones with stage III tumors. “

Reviewer 2 Report

The authors describe here an association of tumor buds and MMP/TIMP expression with survival of breast cancer patient. Tumor budding and it's association with aggressiveness and survival has been described before and while as the authors state the clinical use of this method is limited by the lack of unified methods for quantification, the presented work also has some limitations that should be addressed.

Major concerns

  1. Quantification of tumor buds: The author write in the methods that tumor buds are quantified as the average number of tumor buds in the peritumoral area in those areas with high tumor bud numbers. First of all with the heterogeneity of counted tumor buds would the median be better? And secondly, does this mean that the authors excluded areas with 0 tumor buds? But wouldn't this bias the analysis and should the authors better quantify buds over the complete area? Or at least could the authors normalise the chosen area against the total analysed area.
  2. In relation to the above question: Is the observation that tumor buds in the peritumoral area is highly variable of biological significance. Did the authors stain MMPs in areas with high vs low number of tumor buds and do the see differences within the tumors. Also are there any other histological differences between the areas?
  3. I'm not sure whether the correlation and association analysis in tables 2 and 3 are corrected for multivariate testing? I also would suggest that each table legend includes a description of the statistical method use in addition to the description in the method section.
  4. The authors state that MMPs and TIMPs are expressed in similar levels between cancer cells and tumor buds. Could the authors please provide a quantification as well as clearly labelled representative images?
  5. Am I correct in assuming that MMP and TIMP expression was only quantified as present in all cancer/stromal cells vs. not present in cells? If not what was the cut-off of positive cells that would result in a tumor being considered as positive for each marker? I could not find the exact description of the quantification in the methods.
  6. The abstract and table 3 indicate that TIMP-1 and TIMP-2 were quantified, but the text states that also TIMP-3 was measured. If this is true, please include the data in all analysis.
  7. It is not clear to me what stainings are presented in Figure 3. What marker is shown? A mix of all analysed once only one? I would strongly suggest to show representative images of each marker (MMP or TIMP, clearly labelled), which should show staining in cancer cells, tumor buds, CAFs, and MICs. It also seems to me that different magnifications are presented but I'm missing size bars in the picture or at least an indication of the magnification in the legend. Further could the authors clearly indicate different cell types in the representative images by arrows?
  8. The authors state in the methods that vimentin, cytokeratin, and CD45 were used to differentiate cell types. In how many cases of the cohort? Was this done by multiplex IHC to stain MMPs and TIMPs on the same slide? I would like to see representative images in the manuscript.
  9. The authors suggest that MMP expression in stromal cells is involved in tumor budding? Is this really supported by the data? Could the authors compare MMP expression within the same tumor in areas with or without tumor buds and see if the MMP expression in stromal cells differs if they are close to a tumor bud or in area devoid of tumor buds?

Minor comments:

  1. Figure 6 should be figure 1
  2. In figure 6 the location HPF area should be outlines in the overview picture.

Author Response

Reviewer 2

  1. First of all with the heterogeneity of counted tumor buds would the median be better? And secondly, does this mean that the authors excluded areas with 0 tumor buds? But wouldn't this bias the analysis and should the authors better quantify buds over the complete area? Or at least could the authors normalise the chosen area against the total analysed area.

Following the Reviewer’s comment, we have indicated in the Materials and Methods section (page 3, line 104) that we have chosen the areas with the highest budding density (“hotspots”), such as it was also performed by other authors (references 15, 16 y 17 of the revised version of the manuscript, and also by other works published and not cited such as: Graham et al., Am J Surg Pathol 2015; Miyata et al., Cancer 2009, among others). Also, we have described that at least 10 high power fields (HPF) (0.55 mm) per case were scored and we did not exclude areas without tumors buds (page 3, line 104). In the Results section (page 6, line 176), we have indicated that areas with high number of tumor buds shown a more infiltrative pattern and peritumoral stroma response.

The median value of number of tumor buds was 5, as reported in the first sentence of the Results section. However, as shown in the Figure 3A of the revised version of the manuscript, this value does not improve the value of 4 as the optimal cut-off point for predicting relapse-free survival in our patient population.

Nevertheless, and in agreement with the Reviewer´s comment, we have indicated (Discussion section page 12, line 345) that another limitation of the study was the difficulty of evaluating the heterogeneity of counted tumor budding along the tumor sample. In addition, we consider that further studies will be carried out to explore a more accurate and rapid evaluation of this morphologic finding, like the use of artificial intelligence tools

  1. In relation to the above question: Is the observation that tumor buds in the peritumoral area is highly variable of biological significance. Did the authors stain MMPs in areas with high vs low number of tumor buds and do the see differences within the tumors. Also are there any other histological differences between the areas?

We have evaluated MMPs staining in the whole of tissue array section (1,5 mm in diameter) corresponding to two cores per case, through a histologically representative area of each tumor and without differentiating areas with high vs. low number of tumor buds. We did not find significant differences for MMPs or TIMPs immunostaining between tissue array sections of each tumor or between the different areas of each section. However, we found that areas with high budding had a more infiltrative pattern and peritumoral stroma response. These aspects are now indicated in the revised version of the manuscript (In Materials and Methods section: page 8, line: 223; and in Results section:  page 6, line: 176).

  1. I'm not sure whether the correlation and association analysis in tables 2 and 3 are corrected for multivariate testing? I also would suggest that each table legend includes a description of the statistical method use in addition to the description in the method section.

As suggested by the reviewer, we have indicated in the footnotes to Tables 2 and 3 that the chi-square test was applied for data analysis.

  1. The authors state that MMPs and TIMPs are expressed in similar levels between cancer cells and tumor buds. Could the authors please provide a quantification as well as clearly labelled representative images?

The differences between cancer cells and tumor buds for both MMPs and TIMPs immunostaining were less than 5%, such as it is now reported in the revised version of the manuscript (in Results section:  page 8, lines: 221). The new Figure 4 allows comparison of immunostaining in cancer cells (black arrows) and tumor buds (red arrows) for each protein.

  1. Am I correct in assuming that MMP and TIMP expression was only quantified as present in all cancer/stromal cells vs. not present in cells? If not what was the cut-off of positive cells that would result in a tumor being considered as positive for each marker? I could not find the exact description of the quantification in the methods.

As recommended by the Reviewer, we have explained that “In tumors positive for cells expressing either MMPs or TIMPs, at least 70% of these cells showed a positive immunostaining of each evaluated field”. (Results section: page 8, lines: 215).

  1. The abstract and table 3 indicate that TIMP-1 and TIMP-2 were quantified, but the text states that also TIMP-3 was measured. If this is true, please include the data in all analysis.

Initially, we analyzed the expression of TIMP-3 in a smaller number of tumors; however, the preliminary results do not point to significant data. Therefore, data on TIMP-3 were removed from the revised version of the manuscript.

  1. It is not clear to me what stainings are presented in Figure 3. What marker is shown? A mix of all analysed once only one? I would strongly suggest to show representative images of each marker (MMP or TIMP, clearly labelled), which should show staining in cancer cells, tumor buds, CAFs, and MICs. It also seems to me that different magnifications are presented but I'm missing size bars in the picture or at least an indication of the magnification in the legend. Further could the authors clearly indicate different cell types in the representative images by arrows?

Following the reviewer’s recommendation, we changed the Figure 3 (now named Figure 4) in the revised version of the manuscript. The new Figure 4 shows representative staining of each MMPs/TIMPs by cancer cells, tumor buds, MICs and CAFs. In addition, details on magnification in the legend and arrows indicating the different cell types were also included.

  1. The authors state in the methods that vimentin, cytokeratin, and CD45 were used to differentiate cell types. In how many cases of the cohort? Was this done by multiplex IHC to stain MMPs and TIMPs on the same slide? I would like to see representative images in the manuscript.

Following the Reviewer’s indication, we have indicated in the new version of the manuscript (page 4, line 109) that these methods were used in an exceptional way as morphological criteria were sufficient in most cases.

  1. The authors suggest that MMP expression in stromal cells is involved in tumor budding? Is this really supported by the data? Could the authors compare MMP expression within the same tumor in areas with or without tumor buds and see if the MMP expression in stromal cells differs if they are close to a tumor bud or in area devoid of tumor buds?

Such as we have indicated above in major concern 2, as well as in the text, we did not find significant differences between areas with and without tumor buds. According to the Reviewer´s observation, and despite any significant association between the global expressions of MMPs/TIMPs and budding grade found in our study, we have eliminated in the discussion section the sentences referring to the MMPs expression could be involved in tumor budding. We think that the key input of the present study is improving the prognostic evaluation in breast cancer by integrating a morphological feature, such as tumor budding grading, into the context of the biological phenotype of the stromal cells from the invasive tumor front, such as their MMPs/TIMPs expressions.

Minor comments:

  1. Figure 6 should be figure 1

All figures are now correctly numbered in the revised version of the manuscript.

  1. In figure 6 the location HPF area should be outlined in the overview

Following the Reviewer´s recommendation, the location HPF area was outlined in the overview picture of the new Figure 1.

Round 2

Reviewer 2 Report

The authors have responded sufficiently to the reviewers comments and have improved their manuscript.

2 concerns remain for me:

1) The quality of the images is poor in all figures and needs to be improved before publication. The text in figure 5 is also very small and font should be increased

2) Looking at the references the authors have cited their own work extensively (3 of the authors appear 8,11, and 12 times in a list of 56 references). I suggest that the authors select and keep only the most relevant of these citations in the manuscript.

Author Response

Please find enclosed the revised version of our manuscript (biomedicines-1057716) where we have responded to the Reviewer’s comments. We thank you and reviewers for the valuable criticisms and suggestions.

Our specific answers are listed below.

Reviewer 2

  1. The quality of the images is poor in all figures and needs to be improved before publication. The text in figure 5 is also very small and font should be increased

We have improved the quality of all images and included them in the new version of the manuscript. Also, we have submitted them as pdf (as required) but we can send all figures in tiff format.

  1. Looking at the references the authors have cited their own work extensively (3 of the authors appear 8, 11, and 12 times in a list of 56 references). I suggest that the authors select and keep only the most relevant of these citations in the manuscript.

Following the Reviewer’s comments, we have deleted the following references:

  • Page 2 line 61: Eiro, N.; Fernandez-Garcia, B.; Gonzalez, L.O.; Vizoso, F. Clinical Relevance of Matrix Metalloproteases and their Inhibitors in Breast Cancer. Mutagenesis and carcinogenesis 2013, S13: 004., doi:10.4172/2157-2518.S13-004.
  • Page 5 line 144: Gonzalez, L.O.; Corte, M.D.; Junquera, S.; Gonzalez-Fernandez, R.; del Casar, J.M.; Garcia, C.; Andicoechea, A.; Vazquez, J.; Perez-Fernandez, R.; Vizoso, F.J. Expression and prognostic significance of metalloproteases and their inhibitors in luminal A and basal-like phenotypes of breast carcinoma. Hum Pathol 2009, 40, 1224-1233, doi:10.1016/j.humpath.2008.12.022.

We cannot delete more references because they support data cited in the manuscript:

  • Citations 21 to 24 refer to previous studies where some of these patients were included, as we must to indicate.
  • Reference 28 supports the previous validation of antibodies used.
  • Reference 37 supports the mechanism and molecules secreted by CAFs to contribute to breast cancer progression.
  • References 49 to 51 supports different data regarding inflammatory cells and different gene expression profiles.